# Hypertension, Thrombosis, Kidney Failure, and Diabetes: Is COVID-19 an Endothelial Disease? A Comprehensive Evaluation of Clinical and Basic Evidence

**DOI:** 10.3390/jcm9051417

**Published:** 2020-05-11

**Authors:** Celestino Sardu, Jessica Gambardella, Marco Bruno Morelli, Xujun Wang, Raffaele Marfella, Gaetano Santulli

**Affiliations:** 1Department of Advanced Medical and Surgical Sciences, University of Campania “Luigi Vanvitelli”, 80100 Naples, Italy; drsarducele@gmail.com (C.S.); raffaele.marfella@unicampania.it (R.M.); 2Department of Medical Sciences, International University of Health and Medical Sciences “Saint Camillus”, 00131 Rome, Italy; 3Department of Advanced Biomedical Sciences, International Translational Research and Medical Education Academic Research Unit (ITME), “Federico II” University, 80131 Naples, Italy; jessica.gambardella@einsteinmed.org; 4Department of Medicine, Division of Cardiology, Albert Einstein College of Medicine, Wilf Family Cardiovascular Research Institute, New York, NY 10461, USA; marco.morelli@einstein.yu.edu (M.B.M.); xujun.wang@einsteinmed.org (X.W.); 5Department of Molecular Pharmacology, Fleischer Institute for Diabetes and Metabolism (FIDAM), Montefiore University Hospital, New York, NY 10461, USA

**Keywords:** ACE2, acute kidney injury, blood pressure, catepsin, coronavirus, COVID, cytokine storm, endothelium, heparin, Kawasaki disease

## Abstract

The symptoms most commonly reported by patients affected by coronavirus disease (COVID-19) include cough, fever, and shortness of breath. However, other major events usually observed in COVID-19 patients (e.g., high blood pressure, arterial and venous thromboembolism, kidney disease, neurologic disorders, and diabetes mellitus) indicate that the virus is targeting the endothelium, one of the largest organs in the human body. Herein, we report a systematic and comprehensive evaluation of both clinical and preclinical evidence supporting the hypothesis that the endothelium is a key target organ in COVID-19, providing a mechanistic rationale behind its systemic manifestations.

## 1. Introduction

Coronavirus disease (COVID-19) represents a public health crisis of global proportions [1]. Caused by the *severe acute respiratory syndrome corona virus 2* (SARS-CoV-2), COVID-19 was first announced in December 2019 in Wuhan, the capital of China’s Hubei province [2,3].

The symptoms most commonly reported include cough, fever, and shortness of breath. The pathophysiology of the disease explains why respiratory symptoms are so common: indeed, the virus accesses host cells via the protein angiotensin-converting enzyme 2 (ACE2) [4,5], which is very abundant in the lungs [6].

Nevertheless, ACE2 is also expressed by endothelial cells [7,8], and other major clinical events usually observed in COVID-19 patients (e.g., high blood pressure [9,10,11,12,13], thrombosis [14,15,16] kidney disease [17,18], pulmonary embolism [19,20], cerebrovascular and neurologic disorders [21,22]) indicate that the virus is targeting the endothelium [23], one of the largest organs in the human body [24,25,26]. The cases of Kawasaki disease reported in young COVID-19 patients [27] support our view of a systemic vasculitis caused by SARS-CoV-2.

## 2. Pathogenesis of COVID-19

To access host cells, SARS-CoV-2 uses a surface glycoprotein (peplomer) known as spike; ACE2 has been shown to be a co-receptor for coronavirus entry [28,29,30]. Therefore, the density of ACE2 in each tissue may correlate with the severity of the disease in that tissue [31,32,33,34,35,36]. Other receptors on the surface of human cells have been suggested to mediate the entry of SARS-CoV-2 [5], including transmembrane serine protease 2 (TMPRSS2) [37,38], sialic acid receptors [39,40], and extracellular matrix metalloproteinase inducer (CD147, also known as basigin) [41]. Additionally, catepsin B and L have been shown to be critical entry factors in the pathogenesis of COVID-19 [38,42].

Intriguingly, all of these factors involved in the entry of SARS-CoV-2 in the host cell are known to be expressed by endothelial cells [43,44,45,46,47,48,49] (Figure 1).

ACE2 remains the most studied of these receptors [34,50,51,52,53,54]: for instance, its genetic inactivation has been shown to cause severe lung injury in H5N1-challenged mice [55], whereas administration of recombinant human ACE2 ameliorates H5N1 virus-induced lung injury in mice [55].

ACE2 is currently at the center of a heated debate among physicians [56,57,58,59], and there are concerns that medical management of hypertension, including the use of inhibitors of the renin-angiotensin-aldosterone system (RAAS), may contribute to the adverse health outcomes observed [34,60,61]; TMPRSS2 binds the viral spike glycoprotein [37]; recent structural assays have suggested that coronaviruses can bind sialic acid receptors [39]; CD147 has been shown to be essential for the entry of cytomegalovirus into endothelial cells [46]; both catepsin B [47] and L [49] are present in endothelial cells (Figure 1).

The endothelium prevents blood clotting by providing an antithrombotic surface, maintained by heparan sulphate present in the matrix surrounding the cells [62,63], by the expression of tissue factor inhibitor [64], thrombomodulin [65], and by the production of tissue-type plasminogen activator that promotes fibrinolysis [66,67].

Endothelial dysfunction refers to a systemic condition in which the endothelium loses its physiological properties, including the tendency to promote vasodilation, fibrinolysis, and anti-aggregation [68,69,70,71,72]; moreover, endothelial dysfunction appears to be a consistent finding in patients with diabetes [69,73,74,75,76,77,78]. Here we will discuss clinical and preclinical findings supporting our hypothesis [79] that COVID-19 impairs endothelial function (Figure 1).

## 3. Hypertension and COVID-19

Several investigators have called attention to the potential over-representation of hypertension among patients with COVID-19 [13,80,81,82]. Moreover, hypertension appears to track closely with advancing age, which is emerging as one of the strongest predictors of COVID-19–related death [14,83]. Specifically, observational trials and retrospectives studies conducted near Wuhan area have actually shown that hypertension is the most common co-morbidity observed in patients affected by COVID-19, ranging from 15% to over 30% [14,84,85,86,87].

One of the largest studies has been conducted by Guan et al. between December 11, 2019, and January 29, 2020, providing data on 1099 hospitalized patients and outpatients with laboratory-confirmed COVID-19 infection [84]; in this cohort, 165 (~15%) had high blood pressure [84]. The authors also evaluated the severity of disease, and the composite outcome of intensive care unit (ICU) admission, mechanical ventilation and death, concluding that 23.7% of hypertensive patients had disease severity (vs. 13.4% of normotensive subjects), and that 35.8% (vs. 13.7%) reached the composite endpoint of ICU admission, mechanical ventilation and death [84].

The high rate of hypertensive patients with COVID-19 was later confirmed in a prospective analysis on 41 patients admitted to hospital in Wuhan [85] as well as in a large study conducted on 138 hospitalized patients with confirmed COVID-19 infection [86]. In the latter report, the rate of hypertension was 31.2%, and 58.3% of hypertensive patients with COVID-19 infection were admitted to ICU compared to 21.6% of individuals with normal blood pressure [86], evidencing the hypertensive state as a common co-morbidity and cause of ICU admission in COVID-19 patients [86].

Similarly, among 191 COVID-19 patients from Jinyintan Hospital and Wuhan Pulmonary Hospital, 58 (30%) had hypertension, and 26 (48%) did not survive COVID-19, whereas 32 (23%) were survivors [14]. The 30% rate of hypertensive patients was further confirmed in an analysis based on the severity of COVID-19 conducted on 140 patients in Wuhan: 58 patients were classified as severe vs. 82 patients classified as not severe: hypertensive patients represented 37.9% of severe vs. 24.4% of not severe COVID-19 patients [87]. In a cohort of 1590 patients from 575 hospitals, underlying hypertension was independently associated with severe COVID-19 (hazard ratio 1.58, 95% CI: 1.07–2.32) [13]. Overall, these findings confirm a dual aspect of hypertension during COVID-19 pandemic: first, hypertension is the most common co-morbidity observed in COVID-19 patients; second, hypertension is evidenced in patients with worse prognosis and higher rate of death.

These studies also raise numerous questions regarding the association between hypertension and COVID-19. Indeed, hypertension is known to be one of most common diseases and co-morbidities worldwide, considered a silent killer for the worldwide population [88]. We speculate that the higher rate of hypertension and the worse prognosis in patients with COVID-19 infection could be seen as the spy of a cause-effect mechanism, more than as a casual pre-existing association between these two different diseases.

## 4. ACE2 and Anti-Hypertensive Drugs: What Do We Know?

ACE inhibitors (ACEi) and angiotensin II receptor blockers (ARB) represent very effective strategies for the treatment of hypertension [88]. These drugs reduce the effects of renin-angiotensin axis by inhibiting ACE (ACEi) or by blocking the angiotensin receptors (ARB), as shown in Figure 2. A growing question for the scientific community and physicians is to understand whether ACEi/ARB could affect the prognosis of hypertensive COVID-19 patients [34,89,90,91].

The exact role of ACEi/ARB in the control of ACE2 molecular pathways is controversial: indeed, preclinical studies evidenced that the selective blockade of either angiotensin II synthesis or activity in rats induces increases in ACE2 gene expression and activity [92,93,94,95,96]; similarly, treating infarcted rats with ARB increased plasma concentration of angiotensin 1–7 and ACE2 [97]. In mice, ARB treatment augmented ACE2 mRNA and protein levels [98,99] and prevented the decrease in ACE2 protein levels induced by Angiotensin II [100]. Equally important, mineralocorticoid receptor blockers prevented aldosterone-induced reduction in cardiac ACE2 mRNA expression in rat cardiomyocytes [101] and increased ACE2 expression and activity in murine hearts and in monocyte-derived macrophages obtained from ten patients with heart failure [102].

Nevertheless, there is no clinical evidence that ACEi could directly affect molecular pathways linked to ACE2 activity. For instance, urinary ACE2 levels were reported to be higher in patients treated with olmesartan vs. untreated controls, but this finding was not observed in patients treated with other ARB or enalapril [103]; instead, another study reported no difference in ACE2 activity in patients who were taking ACEi or ARB vs. untreated patients [89]. Of note, ACE2, which functions as a carboxypeptidase [104] is not inhibited by clinically prescribed ACEi.

In particular, ACE2 acts to counterbalance the effect of ACE [105]; indeed, whereas ACE generates angiotensin II from angiotensin I, ACE2 converts angiotensin II into an active heptapeptide (angiotensin 1-7), which binds the Mas receptor (MasR), triggering vasodilative, anti-oxidant, and anti-inflammatory properties [106,107,108,109] (Figure 2).

Some media sources have recently called for the discontinuation of ACEi and ARB, both prophylactically and in the context of suspected COVID-19 [110]. However, several associations have recommended not to suspend these therapies [61,111,112,113,114], and these recommendations have been confirmed by three recent studies: the first one performed on 362 hypertensive patients showed that ACEIs/ARBs are not associated with the severity or mortality of COVID-19 [91]; the second one verified the effects of ACEI/ARB on 1128 hypertensive COVID-19 patients, showing that the use of ACEI/ARB was associated with lower risk of all-cause mortality compared with ACEI/ARB non-users [115]; the third one demonstrated that without increasing the risk for SARS-CoV-2 infection, ACEI/ARB outcompeted other antihypertensive drugs in reducing inflammatory markers like C-reactive protein and procalcitonin levels in COVID-19 patients with preexisting hypertension [116]. Consistent with these findings, three observational studies performed in different populations and with different designs [117,118,119] (published in the same issue of the *New England Journal of Medicine*), arrived at the consistent message that the continued use of ACEI/ARB is unlikely to be harmful in COVID-19 patients. Notably, in one of these studies [117], the use of either ACEI or statins—two classes of drugs that are known to ameliorate endothelial function [120,121,122,123]—was found to be associated with a lower risk of in-hospital death than non-use.

The binding of the SARS-CoV-2 spike protein to ACE2 has been suggested to cause the down-regulation of ACE2 from the cell membrane [124]. Consequently, ACE2 down-regulation could lead to a loss of protective effects exerted by ACEi/ARB in humans [125]. Such down-regulation of ACE2 is an attractive research field [95,126,127,128]. Indeed, it could be a valid therapeutic target to ameliorate response and clinical prognosis in hypertensive patients affected by COVID-19. Moreover, some investigators proposed the restoration of ACE2 by administration of recombinant ACE2 to reverse the lung-injury process during viral infections [4]. Actually, these effects are being investigated in ongoing clinical trials (ClinicalTrials.gov NCT04287686), alongside the use of losartan as first therapy for COVID-19 in hospitalized (NCT04312009) or not hospitalized patients (NCT04311177). A major role in the pathogenesis of (as well as in the clinical response to) COVID-19 could also be played by ACE2 polymorphisms, which are relatively under-investigated if compared to ACE [129,130].

Finally, we have to consider the higher rate of cardiac injury and adverse outcomes in hypertensive patients during the COVID-19 pandemic [131,132,133]. Therefore, ACEi/ARB chronic therapy should not be discontinued in hypertensive patients with COVID-19. Indeed, the loss of their pneumo- and cardio- protective effects could be detrimental [88]. In addition, in the absence of adequate follow-up visits, switching from ACEi/ARB to another anti-hypertensive therapy could cause a suboptimal control of blood pressure.

Thus, as suggested by several medical associations [110], in the absence of definitive clinical studies and without clear evidence, hypertensive patients should avoid discontinuation and/or therapeutic switching during COVID-19 infection.

Another noteworthy feature of COVID-19 for cardiologists is the significant decrease in the rates of hospital admissions for acute coronary syndromes which has been reported both in Italy [134] and US [135] during the COVID-19 outbreak, and despite being initially attributed to reduced air pollution, better adherence to treatment, or absence of occupational stress during lockdown, this phenomenon seems to be most likely due to the fear of going to the hospital and/or seeking medical attention during a pandemic. Unfortunately, the current decline in hospitalization for acute coronary syndromes will trigger an increase in cases of heart failure in the near future.

## 5. Kidney Disease in COVID-19

Acute kidney injury (AKI) has been reported in > 20% of critically ill or deceased COVID-19 patients, a percentage that is consistent in studies from China [136], Italy [137] and United States [10]. It is important to note that AKI, proteinuria, and hematuria have been independently associated with a higher risk of death in COVID-19 patients [138]. Furthermore, in a meta-analysis including 1389 COVID-19 patients [139], the prevalence of underlying chronic kidney disease was significantly more frequent among those with a severe COVID-19 disease (3.3% vs. 0.4%; odds ratio 3.03, 95% CI: 1.09–8.47).

According to immunohistochemistry assays [140], ACE2 seems not to be expressed in renal endothelial cells; however, a study based on single-cell analysis has confirmed the expression of ACE2 and TMPRSS2 in human renal endothelial cells [141], and most recently the presence of viral particles was confirmed by electron microscopy in endothelial cells of the glomerular capillary loops of a COVID-19 patient [142]. Besides, endothelial damage was a common finding in renal histopathological analyses of 26 COVID-19 patients, in the absence of interstitial inflammatory infiltrates [143].

## 6. Diabetes and COVID-19

Diabetes mellitus is a frequent co-morbidity and a cause of worse prognosis in COVID-19 patients [12,144,145,146,147,148]. Indeed, evaluating pneumonia cases of unknown causes reported in Wuhan and in patients with history of exposure to Huanan seafood market before Jan 1, 2020, 20% had diabetes [85]. Similarly, among 1099 COVID-19 patients analyzed by Guan and colleagues, 7.4% had diabetes: this percentage goes up to 16.2% among patients with severe disease (vs. 5.7% in patients with non-severe disease) [84]; furthermore, 35.8% of patients experiencing the composite endpoint of ICU admission, mechanical ventilation and death, had diabetes (vs. 13.7% of patients that did not experience such endpoint) [84]. Data from Italy show that more than two-thirds of COVID-19 patients that did not survive had diabetes [149]. In summary, diabetes is a frequent co-morbidity, a risk factor, and an independent prognostic factor in COVID-19 patients. A strong evidence of the negative effects of diabetes in COVID-19 patients is also corroborated by two meta-analyses [150,151].

The worse prognosis in patients with diabetes and COVID-19 could be attributed to the fact that the pneumonia evolves towards clinical stages more refractory to medical therapies, oxygen administration and mechanical ventilation, with necessity of ICU care. These data have been investigated in a previous study conducted in patients with SARS [152], in which the relationship between a known history of diabetes and fasting plasma glucose (FPG) levels with death and morbidity rate was assessed, showing that the percentage of patients with diabetes was significantly higher in deceased vs. survivors (21.5% vs. 3.9%, *p* < 0.01) [152]. Moreover, diabetic subjects with hypoxemia (SaO_2_ < 93%) had higher FPG levels and FPG was independently associated with an increased hazard ratio of mortality (1.1, 95% CI: 1.0–1.1) and hypoxia (1.1, 95% CI: 1.0–1.1) after controlling for age and gender [152]; the authors concluded that both diabetes (3.0, 95% CI: 1.4–6.3) and FPG > or = 7.0 mmol/l (3.3, 95% CI: 1.4–7.7) were independent predictors of death [152].

In COVID-19 patients, the incidence of diabetes is two times higher in ICU/severe vs. non-ICU/severe cases [151]. Indeed, the diagnosis of diabetes in a cohort of patients with COVID-19 infection evidenced a sub-group of patients with a 2.26-fold higher risk of experiencing adverse disease outcome analyses [150]. Additionally, patients with obesity and/or glucose intolerance seem to be particularly vulnerable to COVID-19 [10,148,153,154]. Unfortunately, no data are hitherto available on anti-diabetic medications and glucose homeostasis in COVID-19 patients. This aspect is really limiting, because diabetes and altered glucose homeostasis during a condition of severe pneumonia with SARS are reported as main factors of worse prognosis and death [152]. COVID-19 could also induce new onset diabetes, by augmenting insulin resistance and/or by a direct action [155] on the islets of Langerhans; supporting this view, previous studies have shown that ACE2 can be a therapeutic target to ameliorate microcirculation in the islets [156], and ACE2 is known to be expressed by pancreatic beta cells [157,158,159,160,161,162].

Moreover, frequent cases of ketoacidosis in COVID-19 patients have been reported [163]. Therefore, the investigation of anti-diabetic medications and glucose homeostasis could be harnessed to evaluate patients with higher risk of experiencing worse prognosis and death by COVID-19. We speculate that the amelioration of glucose homeostasis in diabetic COVID-19 patients by specific hypoglycemic drugs could result in the amelioration of clinical outcomes with death reduction. However, these data are not reported in trials on COVID-19, and they need to be investigated in further studies [164].

## 7. Thromboembolism and COVID-19

Patients with COVID19 often show clotting disorders, with organ dysfunction and coagulopathy, resulting in higher mortality [15,165,166]. Critical data came from the analysis of coagulation tests including prothrombin time (PT), activated partial thromboplastin time (APTT), antithrombin activity (AT), fibrinogen, fibrin degradation product (FDP), and D-dimer, in samples collected on admission and during the hospital stay of COVID-19 patients [167]. Non-survivors had significantly higher D-dimer and FDP levels, and longer PT vs. survivors on admission [167]. Moreover, significant reduction and lowering of fibrinogen and AT levels were observed in non-survivors during late stages of hospitalization, which is compatible with a clinical diagnosis of disseminated intravascular coagulation (DIC) [167,168]. Specifically, among 191 COVID-19 patients seen at two hospitals in Wuhan, D-dimer levels over 1 μg/L at admission predicted an 18-fold increase in odds of dying before discharge [14]. Of note, when DIC is caused by a systemic infection, it features an acute systemic over-inflammatory response, strictly linked to endothelial dysfunction [169].

Most recently a case of a COVID-19 patient with an increase of Factor VIII clotting activity and a massive elevation of von Willebrand Factor (vWF) has been reported [170], further supporting our theory: indeed, vWF can be seen as a marker of endothelial damage, since it is normally stored in Weibel-Palade bodies within endothelial cells [171]. Equally important, angiotensin II level in the plasma of COVID-19 patients was markedly elevated and linearly associated to viral load and lung injury [172]; notably, angiotensin II is known to increase microvascular permeability [173,174], to induce the transcription of tissue factor in endothelial cells [175,176,177], and to activate platelets [178,179,180]. Additionally, angiotensin II can trigger the release of several components of the complement system from endothelial cells [181,182,183,184,185,186,187], further corroborating the key role of endothelium in the pathogenesis of venous and arterial thrombosis in COVID-19 patients [188,189].

A dysregulated immune response, as observed in COVID-19, especially in the late stages of the disease, plays a decisive role in endothelial dysfunction and thrombosis [190,191], and microvascular permeability is crucial in viral infections [192]. Indeed, pulmonary endothelium represent a fundamental barrier between the blood and interstitium and have vital regulatory functions; specifically, endothelial cells represent one-third of the cell population of the lung [193], and pulmonary endothelial damage is considered the hallmark of acute respiratory distress syndrome (ARDS) [194]. Animal models of coronavirus-induced severe ARDS have shown that reduced ACE2 activity and loss of ACE2 in the lungs is mirrored by enhanced vascular permeability, and exacerbated pulmonary edema [108]. The functional role of endothelium in pulmonary disease is also suggested by previous reports [195,196]; for instance, the H3N2 influenza virus has been shown to infect endothelial cells in vitro and to trigger endothelial cell apoptosis, which is known to enhance platelet adhesion [197]: endothelial cell death would cause exposure of the extracellular matrix to circulating blood, favoring platelet binding; similarly, the endothelium has been shown to contribute to the development of severe disease during H5N1 influenza infection [198].

Deep vein thrombosis and/or pulmonary embolism have been previously described in patients with SARS [199,200,201,202] and cases of thrombosis complicating influenza-associated pneumonia have also been reported [203,204,205]. Excessive activation of the immune system in response to pathogens can lead to pathological inflammatory consequences. In the case of highly virulent 1918 and avian H5N1 influenza virus infections, the recruitment of inflammatory leukocytes followed by excessive cytokine responses is considered to be the key contributor to morbidity and mortality of the infection [206,207]. Cytokine storm syndromes (CSS) are a group of disorders representing a variety of inflammatory etiologies with the final result of overwhelming systemic inflammation, hemodynamic instability, multiple organ dysfunction, and potentially death [208,209]. Specifically, macrophage activation syndrome [210] and hemophagocytic lymphohistiocytosis (HLH) [211] represent two clinically similar CSS with an unknown degree of etiopathogenic overlap [208]. The interaction between endothelial and immune cells could play a crucial role in COVID-19, especially in severe cases and in the late stages of the disease [212]. For instance, the cytokine storm might lead to an abrupt deterioration of the inflammatory response and hyper-coagulation; the increased vulnerability of patients with cardiovascular diseases and/or diabetes might therefore simply reflect the impact of the underlying chronic inflammation and its response during SARS-CoV-2 infection. If this is the case, endothelial alterations could just be seen as an epiphenomenon.

However, according to numerous investigators, the inflammatory response observed in COVID-19 patients can be considered mild if compared to the one observed in typical ARDS and in cytokine-release syndrome [212,213,214,215]: indeed, in ARDS patients, levels of interleukin-1β and interleukin-6 have been shown to be 10 to 60 fold higher than in COVID-19 [216,217]. Therefore, other mechanisms have to be involved in order to explain the systemic manifestations reported in COVID-19 patients, and endothelial cells, known orchestrators of cytokine amplification during viral infections [218], seem to be one of the best candidates in this sense. Further supporting our view, catecholamines are considered an essential component of the cytokine release syndrome [215] and we have demonstrated that endothelial cells are able to synthetize and release catecholamines [219].

Acute pulmonary embolism, reported in COVID-19 patients [20,220,221,222], has been shown to be a cause of clinical deterioration in viral pneumonias [205,223]. Endothelial dysfunction is known to be a key determinant in hypertension, thrombosis, and DIC [72,224,225,226,227]. Henceforth, it is important to select COVID-19 patients at higher risk of pulmonary embolism, and practice computed tomography pulmonary angiography for the diagnosis of pulmonary thromboembolism especially in case of significant increase of D-dimer values. Anticoagulation could be a necessary therapy to control and reduce pro-thrombotic events, as well as to prevent pulmonary embolism [228].

## 8. Anticoagulation as a Key Therapy for COVID-19

The clinical course of COVID-19 consists of two main phases: viral infection and immune/inflammatory response (Figure 3), which require distinct therapeutic approaches. Strikingly, several drugs suggested as a potential therapeutic strategy for COVID-19 [229,230,231] have been shown to ameliorate endothelial function, including interleukin 6 (IL-6) receptor antagonists (e.g., tocilizumab [232]), colchicine [233], azithromycin [234], and famotidine [235].

Even the antimalaric agents chloroquine and hydroxychloroquine, initially proposed as a therapy for COVID-19 based on anecdotal data [229,236], have been shown to improve endothelial function [237,238]. If our theory is correct [239], other drugs that might be effective in treating COVID-19 patients through their beneficial effects on endothelial cells include α1 adrenergic receptor blockers (e.g., doxazosin) [240], modulators of Sigma receptors [241,242,243], metformin [244], indomethacin [245], and endothelin receptor antagonists (e.g., bosentan) [246]. However, data from randomized trials confirming the actual efficacy of these drugs are not (yet) available.

As discussed before, COVID-19 infection could cause endothelial dysfunction and a hyper-coagulation state. This condition is aggravated by hypoxia, which augments thrombosis by both increasing blood viscosity and hypoxia-inducible transcription factor-dependent signaling pathway [247]. Consequently, these phenomena could result in pulmonary embolism with occlusion and micro-thrombosis in pulmonary small vessels, as observed in critical COVID-19 patients [248]. Apart from cases of pulmonary embolism, COVID-19 can cause a sepsis-associated DIC, which is defined as “sepsis-induced coagulopathy” (SIC) [169]. Thus, there is an increasing interest in anticoagulant therapy to treat COVID-19 [249].

In a retrospective analysis conducted at Tongji Hospital of Huazhong University of Science and Technology in Wuhan, the authors examined 449 patients affected by severe COVID-19 [228]. The diagnosis of severe COVID-19 disease was made by evidence of respiratory rate ≥ 30 breaths/min, arterial oxygen saturation ≤ 93% at rest and PaO_2_/FiO_2_ ≤ 300 mmHg [228]. In these patients, they reviewed and compared the parameters of coagulation tests and clinical characteristics between survivors and non-survivors to evaluate the effects of heparin therapy [228]: 94 patients received low molecular weight heparin (LMWH, 40–60 mg enoxaparin/day) and 5 received unfractionated heparin (UFH, 10000–15000 U/day), without other anti-coagulants [228]. Heparin therapy significantly reduced mortality in patients with SIC score ≥4 (40.0% vs. 64.2%, *p* < 0.05), but not in those with SIC score < 4 (29.0% vs. 22.6%, *p* > 0.05) [228]. D-dimer, PT, and age were positively, while platelet count was negatively, correlated with 28-day mortality [228]. In addition, stratifying by D-dimer values the study population, the authors reported in heparin non-users a rise of mortality linked to the rising D-dimer, and 20% reduction of mortality for patients under heparin with D-dimer exceeding 3.0 μg/mL [228]. Therefore, heparin treatment appears to be associated with better prognosis in severe COVID-19 patients with coagulopathy. The beneficial effects of heparin-based therapies are also supported by the structural analogies between heparin and heparan-sulphate, which according to some investigators may confer heparin with antiviral properties [250,251,252,253,254]. In absence of contraindications, we suggest the use of enoxaparin 40 mg/day in all COVID-19 patients, to be raised up to 1 mg/kg every 12 h in case of D-dimer > 3.0 μg/mL; apixaban (5 mg every 12 h) could represent a useful alternative.

Of course, the full clinical evaluation of patients with COVID-19 infection cannot leave aside the analysis of laboratory and imaging data. We believe that PT/PTT, fibrinogen, and D-Dimer should be monitored daily and anticoagulation therapy should be recommended for COVID-19 patients when the D-Dimer value is four times higher than the normal upper limit, except for patients with anticoagulant contraindications. The confirmed diagnosis of severe COVID-19 disease in patients with hypercoagulation and organ failure could evidence an early stage of sepsis-induced DIC. On the other hand, anticoagulant may not benefit unselected patients. Consequently, further prospective studies are needed to confirm these findings in COVID-19 patients, also testing other anti-aggregants and anti-coagulants (at different doses).

## Figures and Tables

**Figure 1 jcm-09-01417-f001:**
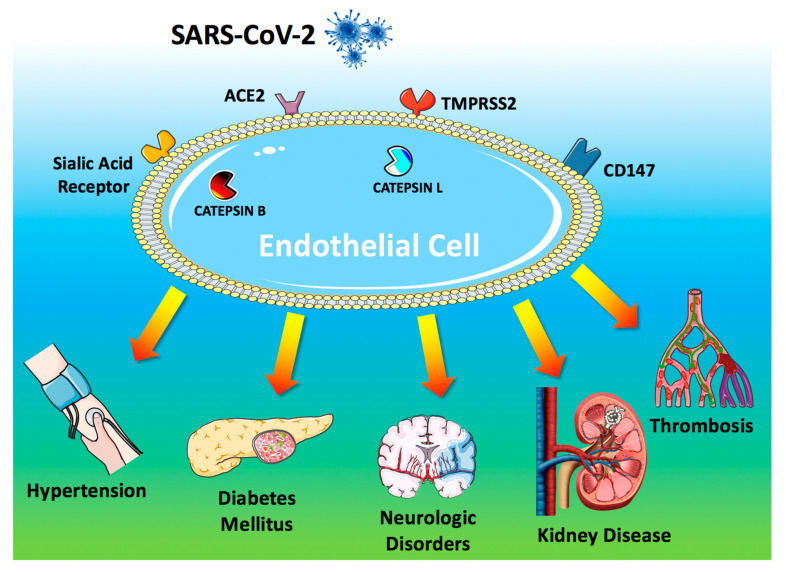
Endothelial dysfunction is a major determinant of COVID-19. The SARS-CoV-2 coronavirus accesses host cells via the binding of its spike glycoprotein to angiotensin-converting enzyme 2 (ACE2), sialic acid receptor, transmembrane serine protease 2 (TMPRSS2), and extracellular matrix metalloproteinase inducer (CD147); catepsin B and L also participate in virus entry. All of these factors are expressed in endothelial cells. Endothelial dysfunction is a common feature of the clinical manifestations observed in COVID-19 patients. All of the drugs proposed as a potential therapeutic strategy to treat COVID-19 patients have been shown to improve endothelial function, including tocilizumab, colchicine, chloroquine/hydroxychloroquine, azithromycin, and famotidine (see text for details and references).

**Figure 2 jcm-09-01417-f002:**
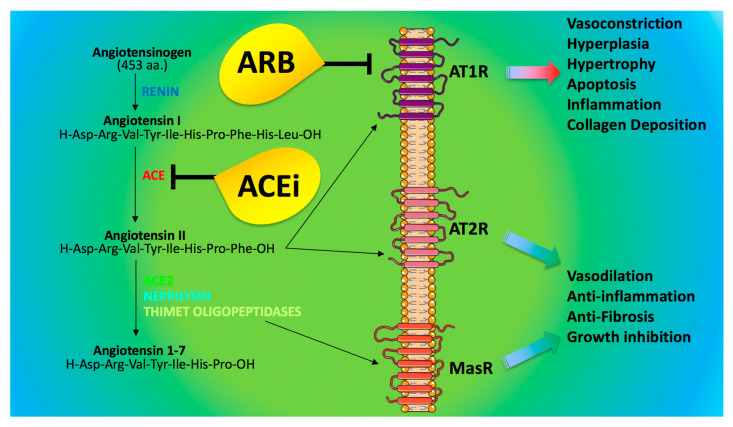
Angiotensin-converting enzyme inhibitors (ACEi) and blockers of the angiotensin receptor 1 (ARB). Angiotensin II and Angiotensin 1–7 binds heptahelical receptors; namely, angiotensin II can activate AT1R (type 1 angiotensin II receptor) and AT2R (type 2 angiotensin II receptor), whereas angiotensin 1–7 binds the Mas Receptor (MasR). The actions mediated by these receptors are depicted in the figure.

**Figure 3 jcm-09-01417-f003:**
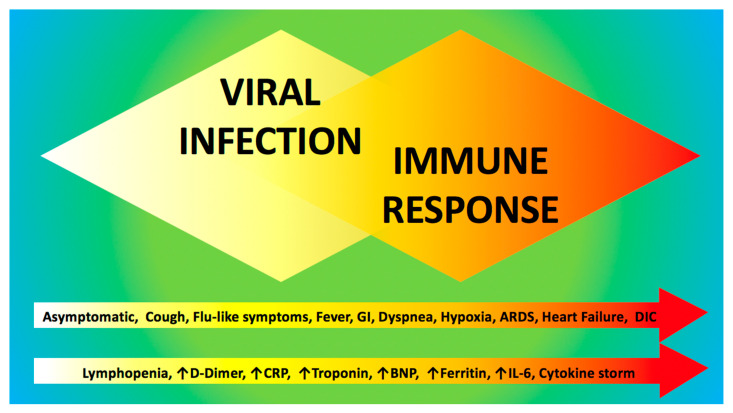
Clinical course of COVID-19 patients. Two main overlapping phases constitute the key pathogenic events in COVID-19: the acute phase represented by viral infection, followed by the immune/inflammatory response. Common clinical and laboratory findings are reported within the arrows at the bottom of the figure.

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
