# Peer review of "Hypertension, Thrombosis, Kidney Failure, and Diabetes: Is COVID-19 an Endothelial Disease? A Comprehensive Evaluation of Clinical and Basic Evidence"

_jcm, 2020, doi:10.3390/jcm9051417_

Round 1
Reviewer 1 Report
In their manuscript the authors are asking themselves (and the readers) if covid-19 is an endothelial disease. This is a very well written essay presenting published data so far regarding potential pathophysiological mechanisms and association with patients suffering from hypertension and DM. Moreover, they further discuss the issue of DIC and hypercoagulation in patients with covid-19.
I however think that every infection with systemic manifestations (unfortunately the term SIRS has been withdrawn from the literature) is involving remote organs through leukocyte-endothelial interactions, as well through cytokines and dangerous signals. Michael Pinsky had characterized sepsis as a 'malignant intravascular inflammation'. In this respect, a dysregulated immune response linked with an unbalanced pro- and anti-inflammatory cytokines' profile is responsible for different clinical phenotypes, depending on virus load, treatment and co-morbidities.
I think that the authors should comment more on this immune-endothelial interaction rather than endothelial per se as a case of severe covid-19. For instance the cytokine storm and the potential HLH syndrome might lead to an abrupt deterioration of the inflammatory response and hyper-coagulation, which always parallels severity of inflammation. The increased vulnerability of patients with cardiovascular diseases or DM might simply reflect the impact of the chronic low grade inflammation and its response during covid-19 infection. In that case, endothelial alterations are the result and not the common denominator.
We still do not know if there is any change in glycocalyx, trans-endothelial tight junctions or expression of different ICAMs at different vascular territories. There are only data for both macro- and microcirculatory thrombosis in pulmonary vessels, but such phenomena could be due to an hyperinflammatory response that has been suggested in these patients.
Finally, DIC that seems to happen is a pre-terminal phenomenon and might be related with a HLH-like syndrome.
Since the authors finally do not answer clearly to the question they raise in the title (there is no conclusions section) i suggest that they should discuss about the differences, if any between covid-19 and other viral pneumonias in terms of systemic inflammatory response, as well as data concerning sepsis in general and propose different studies that should be undertaken in order to evaluate the role of microcirculation and thrombosis in MODS due to covid-19. Finally, they should also comment on the late cytokine storm with DIC and pulmonary-ARDS type alterations that take place in few patients who finally need ICU support with such high mortality.
Author Response
In their manuscript the authors are asking themselves (and the readers) if covid-19 is an endothelial disease. This is a very well written essay presenting published data so far regarding potential pathophysiological mechanisms and association with patients suffering from hypertension and DM. Moreover, they further discuss the issue of DIC and hypercoagulation in patients with covid-19.
I however think that every infection with systemic manifestations (unfortunately the term SIRS has been withdrawn from the literature) is involving remote organs through leukocyte-endothelial interactions, as well through cytokines and dangerous signals. Michael Pinsky had characterized sepsis as a 'malignant intravascular inflammation'. In this respect, a dysregulated immune response linked with an unbalanced pro- and anti-inflammatory cytokines' profile is responsible for different clinical phenotypes, depending on virus load, treatment and co-morbidities.
I think that the authors should comment more on this immune-endothelial interaction rather than endothelial per se as a case of severe covid-19. For instance the cytokine storm and the potential HLH syndrome might lead to an abrupt deterioration of the inflammatory response and hyper-coagulation, which always parallels severity of inflammation. The increased vulnerability of patients with cardiovascular diseases or DM might simply reflect the impact of the chronic low grade inflammation and its response during covid-19 infection. In that case, endothelial alterations are the result and not the common denominator. We still do not know if there is any change in glycocalyx, trans-endothelial tight junctions or expression of different ICAMs at different vascular territories. There are only data for both macro- and microcirculatory thrombosis in pulmonary vessels, but such phenomena could be due to an hyperinflammatory response that has been suggested in these patients. Finally, DIC that seems to happen is a pre-terminal phenomenon and might be related with a HLH-like syndrome. Since the authors finally do not answer clearly to the question they raise in the title (there is no conclusions section) i suggest that they should discuss about the differences, if any between covid-19 and other viral pneumonias in terms of systemic inflammatory response, as well as data concerning sepsis in general and propose different studies that should be undertaken in order to evaluate the role of microcirculation and thrombosis in MODS due to covid-19. Finally, they should also comment on the late cytokine storm with DIC and pulmonary-ARDS type alterations that take place in few patients who finally need ICU support with such high mortality.
We thank this Reviewer for the careful reading of the manuscript, for the words of appreciation toward our review, and for the constructive remarks. We have taken the comments on board to improve and clarify the manuscript. We respect this Reviewer’s opinion on the potential role of immune-endothelial interaction(s), and we now comment on these aspects, as requested.
Few days ago, a Lancet paper has been published showing by electron microscopy viral inclusion structures in endothelial cells of COVID-19 patients (Varga Z, Flammer AJ, Steiger P, Haberecker M, Andermatt R, Zinkernagel AS, Mehra MR, Schuepbach RA, Ruschitzka F, Moch H. Lancet. 2020. pii: S0140-6736(20)30937-5), confirming our theory. The paper is included in our review.
As mentioned above, we have extended our review in order to include most of the topics suggested by this Reviewer. Please note that according to several investigators, the inflammatory response observed in COVID-19 patients can be considered mild if compared to the one observed in typical ARDS and in cytokine-release syndrome (Gattinoni L, et al. Covid-19 Does Not Lead to a "Typical" Acute Respiratory Distress Syndrome. Am J Respir Crit Care Med. 2020; Mehta P et al. COVID-19: consider cytokine storm syndromes and immunosuppression. Lancet. 2020;395:1033-1034; Pedersen SF and Ho YC. SARS-CoV-2: a storm is raging. J Clin Invest. 2020; Staedtke V, Bai RY, Kim K, Darvas M, Davila ML, Riggins GJ, Rothman PB, Papadopoulos N, Kinzler KW, Vogelstein B and Zhou S. Disruption of a self-amplifying catecholamine loop reduces cytokine release syndrome. Nature. 2018;564:273-277): indeed, in ARDS patients, levels of interleukin-1b and interleukin-6 have been shown to be 10 to 60 fold higher than in COVID-19 (Sinha P, Delucchi KL, McAuley DF, O'Kane CM, Matthay MA and Calfee CS. Development and validation of parsimonious algorithms to classify acute respiratory distress syndrome phenotypes: a secondary analysis of randomised controlled trials. Lancet Respir Med. 2020;8:247-257; Qin C, et al. Dysregulation of immune response in patients with COVID-19 in Wuhan, China. Clin Infect Dis. 2020). Therefore, other mechanisms have to be involved in order to explain the systemic manifestations reported in COVID-19 patients.
Reviewer 2 Report
The review by Sardu et al. analysis the pathogenesis of COVID-19 in the context of endothelial dysfunction. The authors discuss several aspects such as hypertension, thrombosis and diabetes and related treatment options. It is a very important review, especially in pandemical situation.
Nevertheless, there are several questions to be answered.
- At the beginning of the review it is important to give a definition of endothelial dysfunction.
- Are there any data about NO bioavailability or endothelial dysfunction markers (ADMA etc.) by COVID-19 patients?
- Are there any autopsy data, that demonstrate the role of endothelial dysfunction and pathological vascular permeability?
- Since ACE2 related information is very important for the treatment, it would be better, if possible, to provide the Figure with Ang-ACE2 signalling and medications in the context of COVID-19.
Minor comment: in the introduction “endothelium one of the largest organs”. Is it an organ?
Author Response
The review by Sardu et al. analysis the pathogenesis of COVID-19 in the context of endothelial dysfunction. The authors discuss several aspects such as hypertension, thrombosis and diabetes and related treatment options. It is a very important review, especially in pandemical situation.
Thank you very much for your comments.
Nevertheless, there are several questions to be answered.
- At the beginning of the review it is important to give a definition of endothelial dysfunction. We provide a definition of endothelial dysfunction in the introduction.
- Are there any data about NO bioavailability or endothelial dysfunction markers (ADMA etc.) by COVID-19 patients? We thank this Reviewer for this pertinent remark. Unfortunately, hitherto there are no peer-reviewed data in this sense, and this is the main reason why there is a question mark in the title of the review. We are currently working on an original paper (with experimental data) to assess endothelial function in COVID-19 patients: our preliminary assays support our theory, but of course they cannot be published in a review manuscript.
- Are there any autopsy data, that demonstrate the role of endothelial dysfunction and pathological vascular permeability? Unfortunately, there are no peer-reviewed data in this sense in COVID-19 patients, and this is the main reason why there is a question mark in the title of the review. However, vascular permeability had been shown in previous coronavirus-mediated lung infections (e.g. SARS: J Pathol. 2015; 235: 185–195).
- Since ACE2 related information is very important for the treatment, it would be better, if possible, to provide the Figure with Ang-ACE2 signalling and medications in the context of COVID-19. We now provide the requested Figure.
Minor comment: in the introduction “endothelium one of the largest organs”. Is it an organ?
The endothelium can be considered an organ; please see below some references:
-Endothelium as an endocrine organ. Annu Rev Physiol; Vol. 57:171-89;
-Endothelium as an organ system. Crit Care Med. 2004;32:271-9;
-Baumgartner-Parzer SM, Waldhäusl WK. The endothelium as a metabolic and endocrine organ: its relation with insulin resistance. Exp Clin Endocrinol Diabetes. 2001.
-Cooke JP. Therapeutic interventions in endothelial dysfunction: endothelium as a target organ. Clin Cardiol. Vol. 20(11 Suppl 2).
Reviewer 3 Report
This is a nice and comprehensive review about some relevant issues of the COVID19.The authors focused on the endothelial cell involvement by the virus. The hypothesis seem interesting and plausible. Well written. Not too much long, which is a good point now that there are so many papers about COVID19.
Author Response
This is a nice and comprehensive review about some relevant issues of the COVID19.The authors focused on the endothelial cell involvement by the virus. The hypothesis seem interesting and plausible. Well written. Not too much long, which is a good point now that there are so many papers about COVID19.
Thank you very much for your comments.
Round 2
Reviewer 1 Report
The authors have included in the revised manuscript suggestions regarding the immune dysregulation and its potential interaction with endothelial cells in a convincing way. Thus, I think that the paper is significantly improved.
Author Response
Thanks!